# The Emotion Regulation Questionnaire: Psychometric Properties and Prediction of Posttraumatic Consequences during the COVID-19 Pandemic in Chilean Adults

**DOI:** 10.3390/ijerph20043452

**Published:** 2023-02-16

**Authors:** Felipe E. García, Pablo Vergara-Barra, Pablo Concha-Ponce, Mariela Andrades, Paulina Rincón, Mauricio Valdivia-Devia

**Affiliations:** 1Departamento de Psiquiatría y Salud Mental, Facultad de Medicina, Universidad de Concepción, Concepción 4070409, Chile; 2Facultad de Psicología, Universidad de Talca, Talca 3465548, Chile; 3Escuela de Psicología y Terapia Ocupacional, Universidad Central de Chile, Santiago 8370292, Chile; 4Departamento de Psicología, Facultad de Ciencias Sociales, Universidad de Concepción, Concepción 4070386, Chile; 5Academia de Ciencias Policiales, Carabineros de Chile, Santiago 7591168, Chile

**Keywords:** coronavirus, posttraumatic stress symptoms, posttraumatic growth, expressive suppression, cognitive reappraisal, predictive validity

## Abstract

The Emotion Regulation Questionnaire (ERQ) is widely used to assess the use of cognitive reappraisal and expressive suppression strategies to regulate negative emotions. The present study evaluates the psychometric properties, reliability and validity of a Chilean adaptation of the ERQ in a large sample of 1543 participants aged between 18 and 87 (38% male, 62% female). The results of the confirmatory factor analysis showed the expected two-factor structure and factorial invariance in relation to gender. Results also indicated adequate internal consistency, test–retest reliability, convergent and predictive validity in predicting posttraumatic stress symptoms and posttraumatic growth six months after the first measurement in a subsample of students exposed to the COVID-19 pandemic. The use of reappraisal was positively associated with general well-being, whereas the use of suppression was positively associated with depressive symptomatology. In terms of posttraumatic consequences, the use of reappraisal was negatively associated with posttraumatic symptomatology and positively associated with posttraumatic growth six months later; in turn, suppression was positively associated with posttraumatic symptomatology and negatively associated with posttraumatic growth six months later. This study demonstrates that the ERQ is a valid and reliable instrument to measure emotional regulation strategies in Chilean adults.

## 1. Introduction

The expression and regulation of emotions are part of everyday life and represent one of the fields of study of psychology. The person affected by a stressful event needs to deal with the emotions, sometimes intense and negative, that the situation generates. To do this, emotional regulation strategies are used [1]; better emotional regulation makes one less vulnerable to developing symptoms of emotional distress or relapse [2,3]. 

The Process Model [4] is one of the most used theories to explain the various emotional regulation strategies. This model defines regulatory strategies as the process by which we influence the emotions we have, when we have them, and the ways in which we experience and express them. It proposes four phases in the generation of these strategies: (a) the modification of the situation that generates the emotion, (b) the attentional deployment to change the focus to another situation, (c) the change in the meaning of the situation, and (d) the modification in the response. Gross and John (2003) highlight two strategies of emotional regulation related to the last two phases mentioned above: cognitive reappraisal and expressive suppression. Cognitive reappraisal is oriented to the change in the meaning of the situation; it involves changing the way we look at the stressful situation to reduce the reaction associated to discomfort. Expressive suppression is oriented to the modification of the response, which consists of the forced attempt not to express the discomfort [5]. 

It has been observed that cognitive reappraisal has a direct relationship with positive indicators, such as optimism and well-being, and an inverse relationship with negative indicators, such as depression; likewise, expressive suppression has a direct relationship with depressive symptoms [6] and an inverse one with optimism and well-being [5]. Likewise, a relationship of cognitive reappraisal with decreased posttraumatic symptomatology (PTSS) [7] and increased posttraumatic growth (PTG) [8,9,10] has been evidenced. In turn, expressive suppression is related to an increase in levels of discomfort, such as perceived stress, depressive symptoms and PTSS [11], which has been observed in people exposed to contagion or lifestyle changes derived from COVID-19 [7,12].

Expressive suppression has been determined to be a strategy used more by men than by women [13,14,15] and cognitive reappraisal has been determined to influence mood more positively in men than in women [16]. In order to measure these two strategies, Gross and John (2003) proposed the Emotional Regulation Questionnaire (ERQ), a 10-item instrument originally built in English. It has since been translated into more than 33 languages, but few studies have formally examined its factorial structure beyond university populations [5,17]. Available studies show the presence of the two original factors: expressive suppression and cognitive reappraisal, and adequate levels of internal consistency in both subscales [18].

The first known translation of the ERQ into Spanish was conducted by Rodríguez-Carvajal et al. (2006); another late version was developed for Peruvian university students by Gargurevich and Matos (2010) [19,20]. Subsequently, Cabello et al. (2013) presented a new translation for Spaniards [21]. Other studies that have evaluated the psychometric properties of the scale in the Spanish language have been Navarro et al. (2018) [22] and Gómez et al. (2016) in Spaniard adolescents [15], Moreta et al. (2021) in Ecuadorian university students [23,24], Olalde et al. (2022) in Mexican adults [25] and Del Valle et al. (2022) in Argentine university students [26]. In all these studies, a factorial structure of two independent factors was obtained; although in a different sample of Ecuadorian university students, Moreta et al. (2022) detected a better fit in a bifactor model composed of a general factor and two specific factors [23]. Internal consistency was adequate for all studies, with the exception of the version by Navarro et al. (2018) where it was unsatisfactory [22]. The study by Gómez et al. (2016) also evaluated test–retest reliability with good indicators [15].

The convergent validity of ERQ in Spanish has been evaluated with other measures of emotional regulation, in addition to negative affectivity and different psychopathological measures [20,21,27]. Regarding measurement invariance, it has been shown to be temporarily invariant, i.e., the items of the ERQ evaluate the same attribute over time [27], and invariant according to sex [15,23,27,28]. Measurement invariance is defined as the probability that an individual obtains a score, showing no change according to the group to which they belong [28]; in this sense, the scores obtained by the ERQ allow comparison over time and between men and women.

Kwon et al. (2013) note that most studies on emotion regulation have been conducted in the United States. However, they claim that each culture encourages and reinforces emotional responses differently and, therefore, they also value differently the emotional regulation strategies used to channel those responses appropriately within a given culture [29]. Liddell and Williams (2019) reinforce the idea that there are cultural differences in the use of these strategies, which could be explained by the more individualistic or more collectivist orientation of each culture [30]. Another axis that could explain the cultural differences within Latin America is the lesser gender differentiation–greater gender differentiation in their rights and in their social roles, placing Chile as a less differentiated country than, for example, Mexico, Ecuador and Argentina [31]. For example, anger is expressed more in Chile than in other Latin American countries [32]. Thus, it is necessary to investigate the psychometric behavior of the scale in each country. In this sense, to date, there is no known study that explores the properties of ERQ in the Chilean population.

Due to its association with emotional discomfort or well-being, in addition to being considered a key concept in the explanation of comorbidity present in various emotional disorders [27], the evaluation of emotional regulation strategies is key to promotion, prevention and intervention in the field of mental health. On the other hand, during the COVID-19 pandemic, emotional distress has increased in the population and, therefore, so did the need to regulate these emotions [7,33]. In a study of the general Chilean population, an increase in depression, anxiety and stress levels was observed during the first months of the pandemic, which the authors attribute to the constant tension caused by the risk of virus infection, changes in lifestyle and quality of life, social isolation, information overload in the media and panic about shortages, among other stressors [33]. In another study with a Chilean university population, it was observed that virtual classes increased stress levels in students, caused by poor internet signal quality, inadequate spaces to connect to classes, with many distractions and a long connection time, among other stressors [7]. This shows the need to have a valid and reliable version of this instrument for use in the adult Chilean population, even more so in a period characterized by the threat and changes in lifestyles caused by the COVID-19 pandemic.

Thus, the aim of this study is to evaluate the psychometric properties of the ERQ in the Chilean adult population. We hypothesize (1) a two-dimensional factorial structure of the ERQ scale; (2) the presence of factorial invariance between women and men; (3) an adequate reliability both in its internal consistency and temporal stability; (4) the existence of convergent validity with depressive symptoms, PTSS, PTG and general well-being; (5) and the existence of predictive validity with PTSS and PTG measured six months after a first assessment in the context of the COVID-19 pandemic. Additionally, gender and age differences are analyzed.

## 2. Materials and Methods

### 2.1. Participants 

One thousand five hundred and forty-three adults participated in this study, a first group of 887 people from the general population, a second group of 147 people who had suffered a stressful event in the last three months, and a third group of 509 people exposed to the changes generated by the COVID-19 health alert, all residents of Chile. In total, 990 were women (64.2%) and 553 were men (35.8%); in the first group, 59.5% were women and 40.5% were men; in the second group, 79.6% were women and 20.4% were men; in the third group, 67.8% were women and 32.2% were men. The overall mean age was 28.82 years (*SD* = 11.24) ranging from 18 to 87 years. The mean age of the first group was 31.93 years (*SD* = 12.36); in the second group, it was 32.95 years (*SD* = 12.22); in the third group, it was 22.20 years (*SD* = 2.87).

Regarding the stressful events to which the participants in second group were exposed, 30.6% were due to social, police, criminal or domestic violence; 19.8% were due to family or partner crisis; 13.6% were due to their own serious illness or that of someone close; 8.2% were due to a work, traffic, or domestic accident; 8.2% were due to the death of someone close to them; and 6.1% were due to work problems.

### 2.2. Instruments 

Emotional regulation. The Emotion Regulation Questionnaire (ERQ) [5] was used. The Spanish translation by Rodríguez-Carvajal et al. (2006) [19] was used, after evaluating its idiomatic relevance, due to its previous use in other research with the Chilean population [6,7,34]. It is composed of ten items divided into two factors: cognitive reappraisal with six items and expressive suppression with four items. It is answered on a Likert scale ranging from 1 (total disagreement) to 7 (total agreement). The psychometric properties in this study are shown in the Results section.

General well-being. The Mental Health Continuum-Short Form (MHC-14) [35] adapted to Spanish by Aragonés et al. (2011) was used [36]. It measures hedonic well-being related to the experience of pleasure and emotions, as well as eudaimonic well-being related to psychological development and personal growth. Together, both measurements establish a general measurement of well-being [37]. The MHC-14 is composed of 14 items that are answered on a 6-point Likert scale, from 1 (never) to 6 (every day). In the present study, the instrument obtained an internal consistency of *α* = 0.88.

Posttraumatic stress symptoms. The SPRINT-E scale [38] was used, translated and validated for Chilean population by Leiva-Bianchi and Gallardo (2013) [39]. It has 12 questions that are answered on a Likert scale from 0 (nothing) to 3 (a lot). The total score was used in the present study. The instrument obtained an internal consistency of *α* = 0.91. 

Posttraumatic growth. The Posttraumatic Growth Inventory, short form (PTGI-SF) [40], in the version translated and validated for the Chilean population by García and Wlodarczyk (2016) was used [41]. It is composed of 10 items that are answered on a 6-point Likert scale, from 0 (no change) to 5 (a very important change). In the present study, the total score was used, and an adequate internal consistency was obtained at *α* = 0.90.

Depressive symptoms. The Center for Epidemiologic Studies Depression Scale (CES-D) [42] translated and validated for Chilean population by Gempp et al. (2004) [43] was used. It is a self-report composed of 20 items that evaluate symptoms associated with depression and in which participants are asked to indicate the frequency at which they experienced each symptom in the most recent week using a 4-point scale ranging from 0 (“rarely or never”) to 3 (“most of the time”). In the present study, the total score was used, and the instrument obtained a high reliability (*α* = 0.85).

### 2.3. Procedure 

Based on the Spanish translation of the ERQ proposed by Rodríguez-Carvajal et al. (2006) [19], the questionnaire was revised following the suggestions of Larrain et al. (2017) [44] for the adaptation of scales. Thus, two psychologist specialists were required to review the items according to three criteria: (a) maintenance of the meaning and intentionality of the statements, (b) use of language appropriate to the context and characteristics of the Chilean population, and (c) review of formal aspects of the instrument. Then, a pilot application was offered to 15 people from the general population: eight men and seven women, with complete primary education, using a saturation criterion. From these reviews, it was assessed that no changes were needed to be made to either the items or the instructions. The scale used can be reviewed in Appendix A.

Participants were selected using three methods. The first group was selected by sampling on successive occasions with a panel sample design. An initial sample of 1198 people out of a total of 1,105,658 inhabitants of the Province of Concepción (Chile), according to the 2017 census data available, were randomly selected and contacted by telephone; of these, 887 people (74.04%) answered the calls and agreed to participate in the study, and were therefore included in the final sample. The scales were applied by previously trained psychologists and psychology students and were answered face-to-face with pencil and paper. Data collection was conducted from August to November 2018. A total of 4% of the data were lost due to incomplete surveys. 

The second group of participants was selected through a general call, through the universities collaborating with the project, from people who had faced a stressful event from a list of previously formulated events, including being a victim of violence, having suffered an accident, and having been exposed to a natural disaster, among others. The event had to have occurred within the last three months and the incentive was the offer of free psychotherapeutic care. Of the 149 people who responded to the call, two were excluded because they were already receiving psychological support for the same event, so 147 people participated. The scores presented in this study were those obtained in the first evaluation, before the interventions. The scales were applied by previously trained psychologists and were answered face-to-face with pencil and paper. Data collection was conducted from March to October 2019. There was no loss of data.

The third group answered a self-administered online questionnaire that was applied to higher education students residing in Chile. Recruitment was conducted through social networks and not in the usual study spaces, since the universities were closed due to the COVID-19 pandemic, so it was a purposive sampling by accessibility. Participants were provided with an access link to a Google form, configured to move to the next item as long as all previous items were answered, so there was no loss of data. In this first measurement, the participants were assigned a code associated with their contact data (telephone and email); they were then contacted six months later so that the same persons could participate in the second assessment. The relationship between the personal code and the contact data was kept confidential by the principal investigator. A total of 509 people were surveyed in October and November 2020 in the first measurement and 502 in April and May 2021 in the second measurement, six months later, as there were seven dropouts. All surveys were answered after signing or accepting an informed consent form in which confidentiality, protection of the data provided, and voluntary participation were guaranteed. 

The whole study was approved by the Ethics Committee of the University of Concepción, in its resolution CEBB 1150-2022.

### 2.4. Data Analysis 

The database was prepared using the Bayesian method for the treatment and replacement of lost values. A multivariate normality analysis was performed through the Mardia coefficient [45]. Normality was confirmed when the values are within the range of ±5 [46]; it is considered appropriate to use the Maximum Likelihood estimation method when the values do not exceed the range of ±70 [47].

To assess the goodness of fit of the two-factor model, a Confirmatory Factor Analysis (CFA) was performed. The *χ*^2^, the Comparative Fit Index (CFI), the Tucker–Lewis Index (TLI), the Root Mean Square Error of Approximation (RMSEA) with its confidence interval, and the Standardized Root mean square (SRMR) were used as fit indices. The criteria for an adequate fit were *χ*^2^ with a *p* > 0.05, the CFI and TLI greater than 0.90, the RMSEA less than 0.08 with a confidence interval not exceeding 0.10, and the SRMR less than 0.08 [48].

Considering the previous findings regarding gender differences in emotional regulation strategies [11], once the factorial structure of the scale was established, a factorial invariance test was performed comparing women and men using successive multi-sample CFA, comparing the nested models with the CFI, TLI and RMSEA adjustment indices. Decreases in the CFI of less than 0.01 (Δ ± 0.01) were considered as an adequate indicator of invariability when compared with the previous model; in addition, a CFI and TLI above 0.90 and an RMSEA below 0.08 were expected [49].

Subsequently, reliability analyses were performed through internal consistency using Cronbach’s *α* and McDonald’s *Ω*, in addition to temporal stability through Pearson’s correlation coefficient *r*. 

For convergent validity, Pearson’s *r* correlation between expressive suppression, cognitive reappraisal, general well-being, depressive symptoms, PTSS and PTG was evaluated. Multiple linear regression was used to evaluate the predictive capacity of cognitive reappraisal and expressive suppression at T1 over PTSS and PTG at T2.

For the analyses, the SPSS v.21 [50] and the AMOS SPSS 20.0 [51] software were used.

## 3. Results

The Mardia coefficient obtained a value of 39.58, so the use of the Maximum Likelihood estimation method in the CFA was selected. As adjustment indices, a *χ*^2^ of 216.89 (*df* = 34), *p* < 0.001, as is usual with large samples, is obtained; however, the other indices show that the two-factor correlated model presents an appropriate fit. These indices are TLI = 0.94; CFI = 0.95; RMSEA = 0.059 (CI: 0.052–0.067); SRMR = 0.041.

Standardized factorial weights can be seen in Figure 1. It can be observed that the factorial weights on the expressive suppression scale varied between 0.58 and 0.83, and on the cognitive reappraisal scale they varied between 0.52 and 0.76. The correlation between expressive suppression and cognitive reappraisal was *r* = 0.20, *p* < 0.001. The modification indices did not suggest any changes that significantly varied the parameters (Figure 1).

Factorial invariance between the samples of women and men was evaluated. To this end, the sequential evaluation of configural, metric, strong and strict invariance was carried out [52]. Configural invariance is the basic model of analysis and requires factors to be specified for the same items in both groups. The refutation of the configural invariance hypothesis implies the absence of substantial equivalence of the construct assessed between women and men. Metric invariance assesses the equality of regression coefficients. Strong invariance assesses equality in intercepts. Strict invariance assesses equality in variance and covariance of errors.

The analysis showed the existence of configural invariance between women and men, as the values of the fit indices were acceptable. Metric invariance was also observed, since the CFI decreased its value by 0.01 compared to the previous model and the TLI and RMSEA were almost unchanged; thus, it can be concluded that the factorial loads are equivalent between women and men. Strong invariance is also accepted as the fit indices are still adequate and the CFI is reduced by only 0.01, so we conclude that the two models evaluated are equivalent with respect to factor coefficients and intercepts. Finally, strict invariance shows no variation in the CFI and the other indices are still adequate, so it is also accepted (see Table 1).

We proceeded to evaluate the internal consistency of the factors. The expressive suppression subscale obtained a *α* Cronbach of 0.76, an *Ω* of 0.76, and the corrected item-total correlation ranged from 0.50 to 0.66. The cognitive reappraisal subscale obtained an *α* of 0.82, an *Ω* of 0.82, with an item-total correlation corrected from 0.46 to 0.67. Regarding the temporal stability of the factors, it is observed that the test–retest correlation in cognitive reappraisal, *r* = 0.69, *p* < 0.001, and expressive suppression, *r* = 0.70, *p* < 0.001, are satisfactory (see Table 2). 

The descriptive statistics of the scales used in the study are presented in Table 2. Skewness and kurtosis indicate the existence of univariate normality in all variables. Through repeated measure ANOVA, it is observed that there are no significant differences in cognitive reappraisal, expressive suppression and PTSS scores between T1 and T2. In contrast, PTG scores at T2 are significantly higher than at T1, *F* (1, 501) = 12.481, *p* < 0.001.

To evaluate convergent validity, the bivariate correlations of cognitive reappraisal and expressive suppression with the other variables measured at T1 were calculated, which is observed in Table 3. Cognitive reappraisal at T1 correlates directly and significantly with general well-being and PTG at T1, and inversely and significantly with depressive and post-traumatic symptoms at T1. In turn, expressive suppression presents a direct and significant correlation with PTSS at T1, but there is no significant relationship with depressive symptoms, general well-being and PTG at T1. It is also observed that the cognitive reappraisal at T1 shows a direct and significant relationship with PTG at T2 and inverse with PTSS at T2. The expressive suppression at T1 shows a direct and significant relationship with PTSS at T2; it shows no significant relationship with PTG at T2 (see Table 3).

In order to evaluate the predictive validity of the emotional regulation strategy, two MLRs were then performed. In each of them, the two types of emotional regulation strategies were included as predictor variables, controlling in a first step the baseline of the predicted criterion variable, either PTSS or PTG at T1. Table 4 shows that expressive suppression directly predicts and cognitive reappraisal inversely predicts PTSS at T2 (*F* = 183.17; *p* < 0.001; *R*^2^ = 0.53). In turn, Table 5 shows that expressive suppression inversely predicts and cognitive reappraisal directly predicts PTG at T2 (*F* = 203.66; *p* < 0.001; *R*^2^ = 0.52) (see Table 4). 

Further analyses assessed whether there was a relationship between regulation strategies with sex, age and participant group. Regarding sex, it is observed that men (*M* = 16.42; *SD* = 5.67) use expressive suppression more than women (*M* = 14.89; *SD* = 5.79), *t*(1535) = −4.970, *p* < 0.001; there are no differences in cognitive reappraisal. In relation to age, a positive correlation is obtained with expressive suppression (*r* = 0.15; *p* < 0.001) and with cognitive reappraisal (*r* = 0.13; *p* < 0.001), that is, the older the age, the more frequent the use of these strategies. Finally, in relation to the group of participants, it is observed that the general population group (*M* = 31.28; *SD* = 7.01) uses the cognitive reappraisal more than the group exposed to a recent stressful event (*M* = 28.94; *SD* = 7.84) or than the group exposed to the changes generated by COVID-19 (*M* = 28.97; *SD* = 6.47) (see Table 5).

## 4. Discussion

Factor analysis confirmed the existence of two correlated factors detected in most of the studies that were analyzed from ERQ: expressive suppression and cognitive reappraisal. The items have positive factorial loads and values higher than 0.52 with each of the dimensions, which is appropriate considering that in psychology factorial loads, the values tend to oscillate around 0.50 [53]. The factorial invariance observed between men and women suggests that the instrument would be free of gender biases, which would allow the comparison of scores between men and women. 

Regarding the reliability of the subscales, the internal consistency indices through Cronbach’s α and McDonald’s Ω are adequate according to the literature [54]. Temporal stability was measured six months after the first evaluation, as suggested by Cohen and Swerdlik (2010); in both subscales, the values obtained were significant and positive [55]. 

Emotional regulation strategies are used to address negative emotions that arise after exposure to stressful events [56]. Expressive suppression is a cognitive avoidance strategy that inhibits the ability to deploy more adaptive resources. Thus, a direct relationship with depressive and post-traumatic symptoms was observed, and inverse relationship was observed with general well-being and PTG. These results coincide with other studies that show the relationship between suppression, depression and hopelessness [57], and it has been shown that their regular use can have a negative impact on mental health [5,32]. It is likely that expressive suppression, by preventing verbal communication about emotions, limits the search for affective support and helps produce feelings of hopelessness [58].

On the other hand, cognitive reappraisal is an emotional regulation strategy that consists of changing the assessment or meaning of the stressful event, which decreases discomfort. Coincidentally, a direct relationship with general well-being and PTG and inverse relationship with depressive and post-traumatic symptoms is observed. In previous studies, a direct relationship with psychological well-being [59] and PTG [60] was observed.

We observed an absence of significant relationship of expressive suppression with general well-being at T1, which contradicts some previous findings [5] but coincides with the findings of Garabito et al. (2020) in a Chilean population [6]. The lack of association between emotional suppression and depressive symptoms is also inconsistent with previous findings [5,6]. It is possible that this lack of association is due to cultural issues such as, for example, a formal education that encourages reading that privileges the cognitive over the emotional and teaches to postpone any current negative effect for the promise of future happiness [61]. 

The ERQ showed the ability to longitudinally predict growth and PTSS associated with lifestyle changes during the COVID-19 pandemic. Expressive suppression directly predicted PTSS and inversely predicted PTG. Other cross-sectional studies during the pandemic also showed a positive relationship of suppression with emotional distress [12]; the present study reinforces this idea by finding the same relationship in a longitudinal assessment. On the other hand, the cognitive reappraisal showed direct prediction of PTG and inverse prediction of PTSS. In another study conducted during the pandemic, reappraisal was associated with a lower level of anxiety in confined people [62]. In this sense, it is possible to consider that the pandemic has generated intense negative emotions in a significant part of the population due to confinement, changes in lifestyle and the direct threat to physical health. Apparently, expressing these emotions instead of repressing them and considering a more acceptable meaning for the changes generated by the pandemic could fulfill a potentially protective function in the face of COVID-19. 

The present study also discovered gender differences, where men showed greater use of expressive suppression compared to women. This result can be explained by the process of socialization and the cultural norms transmitted from childhood that seek to establish social gender roles, which lead boys to suppress or inhibit their emotional expression [63,64] which, according to Walker et al. (2011) [65], would be influenced by the environment, culture, parenting styles and sexual stereotypes to which children are exposed from their proximal environment, as it happens in Chile [66]. A study carried out in Chile by Espinoza and Silva (2014) warned that young Chilean men have been socialized in a cultural system that favors the reproduction of a hegemonic model of masculinity based on a constant comparison and competition between emotional endurance and physical strength [67]. In another study conducted in Chile, it was observed that the formal education of boys and girls excludes readings that promote emotions such as sadness and anger, which could convey the idea that the expression of these emotions is not allowed [61].

In addition to the previous points, it is observed that the older the age, the greater the use of emotional suppression and cognitive reappraisal. In this regard, Urry and Gross (2010) report that as age increases, the emotional regulation strategies that are used around the relationship with others improve [68]. In addition, at an older age, a greater willingness to attend and remember the positive stimuli over the negative ones is observed [69]. Phillips et al. (2006), for their part, add that the most efficient use of these emotional regulation strategies is due to a selective dampening of negative emotions such as anger, a better adaptation of strategies to situations and a more frequent use of positive reevaluation, even in cases experienced as a negative situation [70].

A limitation of this study is the diversity of populations included in the analyses, which in turn occupied different sampling methods, which could alter the overall results presented in this article. For example, only one subsample was evaluated longitudinally, allowing for temporal stability and predictive validity analyses; however, this was an unintentional unrepresentative sample so the generalization of these findings should be taken with caution. The gender disproportion in this study, in which more women than men answered the surveys, is also a limitation. This also affects the generalizability of the results. A third limitation is the absence of data to control some confounding factors for the interpretation of the findings, such as the presence of mental disorders, neurological conditions, and possible infection with COVID-19, among others.

## 5. Conclusions

In conclusion, the ERQ showed a two-dimensional factorial structure, factorial invariance between men and women, adequate internal consistency and temporal stability, convergent validity with depressive symptoms, PTSS, PTG and general well-being, and predictive validity of PTSD and PTG six months after exposure to a stressful event such as the COVID-19 pandemic. In addition, men showed greater use of expressive suppression than women and older participants showed greater use of both strategies than younger ones. These findings contribute to the understanding of the phenomenon of emotional regulation in general and in the Chilean population in particular; for example, the verification of its predictive capacity of mental health consequences would allow its use as a screening instrument to detect more vulnerable people after exposure to a stressful event. The adequate psychometric performance of the scale allows us to conclude that it can continue to be used to evaluate emotional regulation strategies in people exposed to a highly stressful event in this culture.

## Figures and Tables

**Figure 1 ijerph-20-03452-f001:**
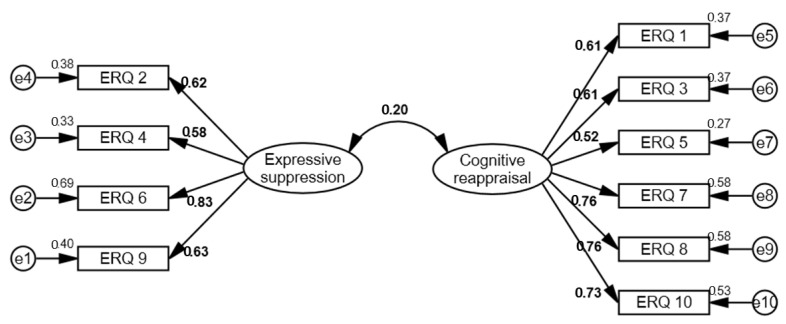
Model of two correlated factors; the values are standardized.

**Table 1 ijerph-20-03452-t001:** Factorial invariance models between women (*n* = 990) and men (*n* = 553).

Models:	*χ* ^2^	Δ*χ*^2^	CFI	ΔCFI	TLI	RMSEA
M1: Configural	2087.59		0.96		0.94	0.045
M2: Metric	2181.08	−93.49	0.95	−0.01	0.94	0.043
M3: Strong	2735.50	−743.52	0.94	−0.01	0.94	0.045
M4: Strict	2924.60	−5660.10	0.94	0.00	0.94	0.044

M1 = not constrained; M2 = M1 + invariant factor loadings; M3 = M2 + invariant intercepts; M4 = M3 + invariant error variances and covariances. Note: CFI comparative fit index, TLI Tucker–Lewis index, RMSEA root mean square error of approximation.

**Table 2 ijerph-20-03452-t002:** Descriptive statistics of the study variables.

Variable	Min	Max	*M*	*SD*	Skewness	Kurtosis
General well-being T1	22	84	62.52	11.31	−0.54	0.15
Depressive symptoms T1	4	57	32.92	11.28	−0.19	−0.34
Posttraumatic stress symptoms T1	5	48	26.66	8.06	0.21	−0.44
Posttraumatic growth T1	0	60	28.72	12.69	0.33	−0.60
ERQ—reappraisal T1	6	42	30.29	7.01	−0.64	0.36
ERQ—suppression T1	4	28	15.44	5.79	−0.01	−0.62
Posttraumatic stress symptoms T2	12	47	28.06	8.19	0.10	−0.65
Posttraumatic growth T2	10	60	32.13	11.93	0.15	−0.85
ERQ—reappraisal T2	6	42	28.90	6.49	−0.67	0.93
ERQ—suppression T2	4	28	15.34	5.66	−0.06	−0.76

ERQ = Emotion regulation questionnaire.

**Table 3 ijerph-20-03452-t003:** Bivariate correlations between emotional regulation strategies and other variables.

Variable	ERQ Reappraisal	ERQ Suppression
ERQ—reappraisal T2	0.69 ***	0.01
ERQ—suppression T2	−0.02	0.70 ***
General well-being T1	0.21 ***	0.01
Depressive symptoms T1	−0.26 **	0.11
Posttraumatic stress symptoms T1	−0.20 ***	0.26 ***
Posttraumatic growth T1	0.31 ***	−0.06
Posttraumatic stress symptoms T2	−0.22 ***	0.25 ***
Posttraumatic growth T2	0.23 ***	−0.14 ***

** *p* < 0.01; *** *p* < 0.001.

**Table 4 ijerph-20-03452-t004:** Multiple linear regression on posttraumatic stress symptoms in T2 (*n* = 509).

Model	Variable	Unstandardized Coefficients		Typified Coefficients	*t*-Value	*p*-Value
		*B*	*SE*	*β*		
1	(Constant)	8.366	0.932		8.979	<0.001
	Posttraumatic stress symptoms	0.708	0.032	0.70	22.019	<0.001
2	(Constant)	11.735	1.641		7.151	<0.001
	Posttraumatic stress symptoms	0.652	0.033	0.65	19.959	<0.001
	ERQ—reappraisal	−0.166	0.040	−0.13	−4.184	<0.001
	ERQ—suppression	0.194	0.046	0.13	4.212	<0.000

**Table 5 ijerph-20-03452-t005:** Multiple linear regression on posttraumatic growth in T2 (*n* = 509).

Model	Variable	Unstandardized Coefficients		Typified Coefficients	*t*-Value	*p*-Value
		*B*	*SE*	*β*		
1	(Constant)	11.153	0.980		11.378	<0.001
	Posttraumatic growth	0.684	0.030	0.72	23.111	<0.001
2	(Constant)	9.092	1.984		4.582	<0.001
	Posttraumatic growth	0.631	0.030	0.66	21.028	<0.001
	ERQ—reappraisal	0.268	0.058	0.15	4.875	<0.001
	ERQ—suppression	−0.267	0.064	−0.13	−4.186	<0.000

ERQ = Emotion regulation questionnaire.

## Data Availability

The raw data supporting the conclusions of this article will be made available by the authors, without undue reservation.

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
