# Peer review of "The Emotion Regulation Questionnaire: Psychometric Properties and Prediction of Posttraumatic Consequences during the COVID-19 Pandemic in Chilean Adults"

_ijerph, 2023, doi:10.3390/ijerph20043452_

Round 1
Reviewer 1 Report
The Emotion Regulation Questionnaire: Psychometric properties and prediction of posttraumatic consequences during the COVID-19 pandemic in Chilean adults
General remarks
The study evaluates the psychometric properties, reliability and validity of a Chilean adaptation of the Emotion Regulation Questionnaire (ERQ) in a large sample of 1543 participants aged between 18 and 87 (38% male, 62% female) consist of three subsamples. The topic is worth of investigation, because the findings could help the understanding of the phenomenon of emotional regulation in the Chilean population and incorporate ERQ in the evaluation of the emotional regulations: cognitive reappraisal and expressive suppression. In people exposed to a stressful events in this culture. I comment on several of instances below.
I suggest the authors do a job on before the editorial process.
Specific remarks
Introduction
Line 101-104: Please clarify the meaning of this sentence ..”Another axis that could explain the cultural differences within Latin America is the less gender differentiation - greater gender differentiation, placing Chile as a less differentiated country than, for example, Mexico, Ecuador and Argentina [31].”
Line 118: The name of scale is missing….. We hypothesize: 1) a two-dimensional factorial structure of the ERQ scale;
Line 121: Put “5” instead “4”….5) predictive validity with PTSS and PTG measured six months after a first assessment, in the context of the COVID-19 pandemic.
Materials and Methods
Participants
Line 126-135:.. Please provide mean age±SD for each subsamples.
Procedure
Line 168-170: In the sentence please provide the name of scale (ERQ?)… „Based on the Spanish translation of the scale proposed by Rodríguez-Carvajal et al. (2006) [19], the questionnaire was revised following the suggestions of Larrain et al. (2017) [44] for the adaptation of scales“.
Please provide how many male and female were included in each subgroups, as well as mean age±SD for each subsamples. How instruments were applied in each subgroups and who applied them?
Please provide more details regarding random selection of the participants in relation to gender, age, and education, as well as the number of adult population of the Province of Concepción (Chile) from available census data (provide year??') in the first group. Please explain why 4% of data in first group were lost?
More information are needed for the second group. How the group was selected? What does mean „partner university institutions“?
The same is for the third group. More information are needed for this group, as well. How the group was selected? What means that „they were closed due to the COVID-19 pandemic.“ Does it mean that they had COVID-19 or not, or something else? Was the second evaluation six months later done for the same person or not and how? Was it possible to assess the same person during the second evaluation because the recruitment was done via Google link? Were there any dropouts in the second evaluation?
Discussion
Line 377-378: This sentence in not clear….. “In addition to what has been stated in the previous points, a direct correlation is observed between people’s age and the use of emotional regulation strategies”. Please provide the direction of age correlation in your finding. Does it mean that younger or older people had better or worse emotional regulation strategies?
Limitations
Possible limitations of gender disproportion should be discussed, as well as the strength. The eligibility criteria needs to be refined. I'd highly suggest the use of a detailed section for the eligibility criteria. For example, the participant did not have any disorder (references), neurological condition (references), any other aspect that affects development (references) were infected in COVID-19, etc. Mostly considering some confounding factors can be a hindrance if not controlled.
Conclusion
The main findings, the novelty of the research and the scientific contribution of the research are not highlighted in the conclusion. Please keep it simple, sharp and focused and connect the findings with the aim of the study, the hypotheses, gender, and age differences.
I recommend revision of the manuscript.
Author Response
Dear reviewer, first of all we would like to thank you for the suggestions given to our manuscript, which will allow us to improve what we have submitted, regardless of your final evaluation. You will find each suggestion below followed by your response (in red)
Introduction
Line 101-104: Please clarify the meaning of this sentence ..”Another axis that could explain the cultural differences within Latin America is the less gender differentiation - greater gender differentiation, placing Chile as a less differentiated country than, for example, Mexico, Ecuador and Argentina [31].”
R: We change the writing: “Another axis that could explain the cultural differences within Latin America is the less gender differentiation - greater gender differentiation in their rights and in their social roles, placing Chile as a less differentiated country than, for example, Mexico, Ecuador and Argentina [31]”.
Line 118: The name of scale is missing….. We hypothesize: 1) a two-dimensional factorial structure of the ERQ scale;
R. Done
Line 121: Put “5” instead “4”….5) predictive validity with PTSS and PTG measured six months after a first assessment, in the context of the COVID-19 pandemic.
R. Done
Materials and Methods
Participants
Line 126-135:.. Please provide mean age±SD for each subsamples.
R. We have now included this information: “The mean age of the first group was 31.93 years (SD=12.36); in the second group it was 32.95 years (SD=12.22); in the third group it was 22.20 years (SD=2.87)”.
Procedure
Line 168-170: In the sentence please provide the name of scale (ERQ?)… „Based on the Spanish translation of the scale proposed by Rodríguez-Carvajal et al. (2006) [19], the questionnaire was revised following the suggestions of Larrain et al. (2017) [44] for the adaptation of scales“.
R. Done
Please provide how many male and female were included in each subgroups, as well as mean age±SD for each subsamples. How instruments were applied in each subgroups and who applied them?
R. The proportion of men and women for each subsample is now included in the participants section. In the procedure section, we now describe in more detail how and by whom the instruments were applied.
In participants: “Of the total, 990 were women (64.2%) and 553 men (35.8%); in the first group 59.5% were women and 40.5% were men; in the second group 79.6% were women and 20.4% were men; in the third group 67.8% were women and 32.2% were men”
Please provide more details regarding random selection of the participants in relation to gender, age, and education, as well as the number of adult population of the Province of Concepción (Chile) from available census data (provide year??') in the first group. Please explain why 4% of data in first group were lost?
R. The procedures section was rewritten to explain in more detail what has been requested by both reviewers.
More information are needed for the second group. How the group was selected? What does mean „partner university institutions“?
R. We have provided more details in the procedure section. In any case, these are partner universities of the project. These universities have care clinics and agreements with different local institutions for patient referral. The call to refer people who have experienced a recent stressful event for psychological support was made by these universities: Universidad Católica del Norte, Universidad Santo Tomás and Universidad de Concepción, in Chile.
The same is for the third group. More information are needed for this group, as well. How the group was selected? What means that „they were closed due to the COVID-19 pandemic.“ Does it mean that they had COVID-19 or not, or something else? Was the second evaluation six months later done for the same person or not and how? Was it possible to assess the same person during the second evaluation because the recruitment was done via Google link? Were there any dropouts in the second evaluation?
R. We clarified in the procedure section that it was the universities that were closed due to the pandemic.
Discussion
Line 377-378: This sentence in not clear….. “In addition to what has been stated in the previous points, a direct correlation is observed between people’s age and the use of emotional regulation strategies”. Please provide the direction of age correlation in your finding. Does it mean that younger or older people had better or worse emotional regulation strategies?
R. We have corrected it: “In addition to the previous points, it is observed that the older the age, the greater the use of emotional suppression and cognitive reappraisal”.
Limitations
Possible limitations of gender disproportion should be discussed, as well as the strength. The eligibility criteria needs to be refined. I'd highly suggest the use of a detailed section for the eligibility criteria. For example, the participant did not have any disorder (references), neurological condition (references), any other aspect that affects development (references) were infected in COVID-19, etc. Mostly considering some confounding factors can be a hindrance if not controlled.
R. The aspects indicated above have now been included in the limitations
Conclusion
The main findings, the novelty of the research and the scientific contribution of the research are not highlighted in the conclusion. Please keep it simple, sharp and focused and connect the findings with the aim of the study, the hypotheses, gender, and age differences.
R. The conclusion has been rewritten
Reviewer 2 Report
OVERALL
- Well-written and presented study.
- The title presumes that the COVID-19 pandemic would be addressed more than it is, other than this is the time period of the study. It is recommended that the authors other reference it and discuss it more throughout the paper, or leave it out of the title and note that this was the timeframe of the study.
INTRODUCTION:
- Abrupt first sentence. Would consider a gentler introductory sentence.
- Paragraphs 2 and 3: I had to reread a few times to see the connection between process model and the next paragraph. May need to make this more clear.
- Please reword the first part of the sentence from lines 64-65 as it is unclear currently.
- Line 74: needs a connecting word before "second"
- Lines 73-79: unclear why there was a distinction between first and second but then just a listing of all spanish-translated iterations. Could be better organized/communicated.
- Given that the COVID-19 pandemic is a main component of the paper, I would attribute more than one line to the impact of the pandemic on emotional regulation.
- Lines 117-123: The aims could be benefited from an English-speaker's edits given nuances in grammar that are incorrect.
MATERIAL AND METHODS:
- The first sentence (lines 126-129) could use English-speaker edits.
- How were participants recruited? (especially those who suffered a "stressful life event") How was their stressful life event determined/categorized?
- Line 159: I think the last sentence is missing a word or two.
- Line 178: Change "A first group" to "The first group"
- Line 184: How were 4% data lost? Didn't answer their phones? Impressive number, just not sure what it means.
- Line 192: Change "A third group" to "The third group"
Results - the results section is very well-written. No changes!
Discussion
- Line 344 - add "a" before "Chilean population"
- Line 346-348: Unclear why a formal education would change the expression of unpleasant emotions?
- Line 349: I would change "the instrument" to the actual name of the instrument
Line 351-353: Needs to be reworded for better grammar
Author Response
Dear reviewer, first of all we would like to thank you for the suggestions given to our manuscript, which will allow us to improve what we have submitted, regardless of your final evaluation. You will find each suggestion below followed by your response (in red).
- The title presumes that the COVID-19 pandemic would be addressed more than it is, other than this is the time period of the study. It is recommended that the authors other reference it and discuss it more throughout the paper, or leave it out of the title and note that this was the timeframe of the study.
R. We have added more detailed information on the impact of COVID-19 on emotions. We believe that the prediction of psychological consequences caused by the pandemic is one of the most important contributions of the present study and therefore prefer to keep it in the title.
INTRODUCTION:
- Abrupt first sentence. Would consider a gentler introductory sentence.
R. We add an introductory sentence: “The expression and regulation of emotions are part of everyday life and represent one of the fields of study of psychology”
- Paragraphs 2 and 3: I had to reread a few times to see the connection between process model and the next paragraph. May need to make this more clear.
R. We have modified it by joining the two paragraphs and clarifying how the strategies proposed by Gross and John relate to the model presented.
- Please reword the first part of the sentence from lines 64-65 as it is unclear currently.
R. The paragraph has been modified
- Line 74: needs a connecting word before "second"
R. Done
- Lines 73-79: unclear why there was a distinction between first and second but then just a listing of all spanish-translated iterations. Could be better organized/communicated.
R. We modified the translation. In this case the word "Spanish" is used interchangeably as a noun referring to the inhabitants of Spain and as an adjective referring to the language used in almost all of Latin America. The indistinct use generates confusion, so the noun is modified to "Spaniard".
- Given that the COVID-19 pandemic is a main component of the paper, I would attribute more than one line to the impact of the pandemic on emotional regulation.
R. Now we add the following paragraph: “In a study of the general Chilean population, an increase in depression, anxiety and stress levels was observed during the first months of the pandemic, which the authors attribute to the constant tension caused by the risk of virus infection, changes in lifestyle and quality of life, social isolation, information overload in the media and panic about shortages, among other stressors (33). In another study with a Chilean university pop-ulation, it was observed that virtual classes increased stress levels in students, caused by poor internet signal quality, inadequate spaces to connect to classes, with many dis-tractions and a long connection time, among other stressors (7)”
- Lines 117-123: The aims could be benefited from an English-speaker's edits given nuances in grammar that are incorrect.
R. We modify the paragraph
MATERIAL AND METHODS:
- The first sentence (lines 126-129) could use English-speaker edits.
R. Done
- How were participants recruited? (especially those who suffered a "stressful life event") How was their stressful life event determined/categorized?
R. The procedure section was rewritten. More details are now provided.
- Line 159: I think the last sentence is missing a word or two.
R. We have corrected it
- Line 178: Change "A first group" to "The first group"
R. We have corrected it
- Line 184: How were 4% data lost? Didn't answer their phones? Impressive number, just not sure what it means.
R. We now explain that the reason was incomplete surveys.
- Line 192: Change "A third group" to "The third group"
R. Done.
Results - the results section is very well-written. No changes!
R. Thank you very much
Discussion
- Line 344 - add "a" before "Chilean population"
R. Done
- Line 346-348: Unclear why a formal education would change the expression of unpleasant emotions?
R. We have changed it to the following sentence: “It is possible that this lack of association is due to cultural issues, for example, a formal education that encourages reading that privileges the cognitive over the emotional and teaches to postpone any current negative affect for the promise of future happiness [61]”.
In any case, the above-mentioned quotation provides details on how the ministerial guidelines for the promotion of reading in Chile favor this type of content.
- Line 349: I would change "the instrument" to the actual name of the instrument
R. Done
Line 351-353: Needs to be reworded for better gramar
R. We modify the writing
Round 2
Reviewer 1 Report
The authors have responded satisfactorily to my comments and suggestions given in my first review. The paper has been much improved after this revision. Only one minor correction remains.
Line 68: Please correct the name of the Emotional Regulation Scale to the Emotional Regulation Questionnaire. Please put the abbreviated name of the questionnaire in parentheses because it is mentioned for the first time in the introduction, e.g. the Emotional Regulation Questionnaire (ERQ).
Author Response
Line 68: Please correct the name of the Emotional Regulation Scale to the Emotional Regulation Questionnaire. Please put the abbreviated name of the questionnaire in parentheses because it is mentioned for the first time in the introduction, e.g. the Emotional Regulation Questionnaire (ERQ).
R. This has been corrected, thank you for pointing out the error.